# Trends in Microbial Community Composition and Function by Soil Depth

**DOI:** 10.3390/microorganisms10030540

**Published:** 2022-02-28

**Authors:** Dan Naylor, Ryan McClure, Janet Jansson

**Affiliations:** Biological Sciences Division, Earth and Biological Sciences Directorate, Pacific Northwest National Laboratory, Richland, WA 99352, USA; ryan.mcclure@pnnl.gov (R.M.); janet.jansson@pnnl.gov (J.J.)

**Keywords:** soil microbiome, soil depth, carbon cycling, soil chemistry

## Abstract

Microbial communities play important roles in soil health, contributing to processes such as the turnover of organic matter and nutrient cycling. As soil edaphic properties such as chemical composition and physical structure change from surface layers to deeper ones, the soil microbiome similarly exhibits substantial variability with depth, with respect to both community composition and functional profiles. However, soil microbiome studies often neglect deeper soils, instead focusing on the top layer of soil. Here, we provide a synthesis on how the soil and its resident microbiome change with depth. We touch upon soil physicochemical properties, microbial diversity, composition, and functional profiles, with a special emphasis on carbon cycling. In doing so, we seek to highlight the importance of incorporating analyses of deeper soils in soil studies.

## 1. Introduction

The microbiome is an essential component of the soil ecosystem, influencing processes such as soil formation and fertility, plant growth and stress tolerance, nutrient turnover, and carbon storage. However, our understanding of the soil microbiome is largely derived from studies conducted on surface layers. Consequently, deeper soil layers represent a historically underappreciated aspect of the soil microbiome. These deeper layers are commonly referred to as “subsoils” and are defined as soil beneath the surface layer. Subsoils have a common bottom cutoff at 30 cm [1,2,3,4], although thresholds from 20 cm [5] to 80 cm [6,7] deep have been used when collecting subsoil samples. Recently, studies have begun to address subsoils as entities distinct from the surface, finding that environmental influences [8], inputs of fresh organic matter (OM) [4], nutrient levels [9], gas and moisture levels [10], and physical structure [11,12] all vary as a function of soil depth. Such changes in soil properties in turn influence the composition of soil microbial communities [13,14,15,16].

Microbial functional profiles will also be affected. Overall activity generally decreases with depth [17], with significant shifts in the functional potential of the microbial community [10,18,19]. Several microbial-mediated soil biogeochemical processes are measurably different by soil layer; these include the cycling of carbon (C), nitrogen (N), phosphorus (P), and sulfur [20,21,22]. For example, at deeper soil levels, overall C cycling rates decline, with longer C residence durations [23] and turnover rates [24]. Therefore, elucidating how microbiome functions vary with soil depth will not only increase our understanding of ecosystem processes, but also improve the accuracy of models of terrestrial C flux [25].

In this review, we explore the influence of soil depth on the soil microbiome, specifically focusing on differences between surface and higher subsoil layers. Deep subsurface soils are not included in this review. We synthesize findings from the literature to address how depth influences soil properties and in turn affects microbial community compositions and functions. We discuss how microbial gene expression patterns and functional guilds are impacted by soil depth, with an emphasis on C cycling. We conclude by discussing the current limitations of soil models and how these limitations may be alleviated in future experiments.

## 2. Soil Biogeochemical Properties and Depth

Soils exhibit heterogeneity in chemical composition and physical structure [26,27], both of which impact microbial community structures [8,28,29]. Indeed, abundances of individual taxa correlate with levels of one or more specific soil factors (e.g., salinity, soil conductivity, pH, mineral composition, C or N contents [22,30,31,32,33]). Here, we review commonly observed shifts in soil mineral and physical characteristics with depth.

### 2.1. Soil Biogeochemical Properties

Soils are characterized by vertical gradients in minerals, including calcium, magnesium, aluminum, iron, and phosphate. Processes that contribute to soil mineral content include weathering (degradation of rocks into smaller minerals), atmospheric deposition [34], leaching (dissolving/transport of nutrients via percolation), root and/or fungal extraction and deposition [35,36], and biological cycling [9], with situational contributions from other processes such as cryoturbation [37] or bioturbation [38]. At the surface, minerals are added through weathering or atmospheric deposition. Then, they are mobilized deeper in the soil profile through processes such as leaching. However, plants will tend to enrich for certain limiting minerals such as P and potassium (K); as such, the levels of these minerals tend to be higher in surface soils where root biomass is at its highest [13]. Meanwhile, less-limiting minerals (sodium, chloride, magnesium) are more subject to leaching and thus their concentrations typically increase with depth [9,39]. Furthermore, the decreased influence of precipitation events in subsoils [40] reduces weathering in those layers, exacerbating mineral accumulation at depth [13].

The local environment (weather patterns and altitude) and soil characteristics (density, texture, and clay/silt/loam content) also play a role in soil chemical gradients. For instance, upon comparison of three soil types with distinct characteristics (alpine steppe, meadow, and swamp meadow), the swamp meadow displayed less pronounced shifts in nutrient contents (C, N, P, K) over depth compared to the former two environments [39]. These findings were speculated to be attributable to a greater influence of cryoturbation and poorer drainage in the swamp meadow [39]. Similarly, a comparative analysis of soils under multiple land-use types (croplands, orchards, abandoned land, and recovered grasslands) found that land-use type, depth, and the interaction effect between these two factors were all significant in explaining variations in physicochemical properties and N content [26]. In a comprehensive study of >10,000 soil profiles, significant differences in depth gradients for nutrients (including C, N, K, calcium, magnesium, among others) were found between the three major soil orders examined [9]. These studies confirm that soil type and the surrounding environment are significant determinants in how biochemical properties change over depth.

### 2.2. Soil Biogeochemical Properties and the Microbiome

Changes in soil abiotic properties can directly or indirectly impact the soil microbiome [22,31,41]. Soil abiotic properties that not only change with depth, but also have measurable effects on the soil microbiome, include P, K, pH, calcium, magnesium, and nitrate contents [13,22,29,30,31]. For example, in a study of soil microbiome depth trends across a reforestation chronosequence, several microbial phyla that varied in abundance by depth displayed significant correlations with the levels of one or more edaphic variables (nitrate, pH, organic matter content, etc.) [22]. Several of these correlations were proposed to be linked to nutrient cycling, as the accumulation of a particular nutrient at a given depth can enrich for microbial communities with the corresponding cycling pathways [22].

In a specific example, calcium promotes the formation of soil aggregates [42] by facilitating the cation bridging of humic acid to clay molecules [43], and this effect is generally more pronounced in deeper soil layers. As soil aggregates contain a variety of metabolic niches [44] that microbes can take advantage of, calcium indirectly increases microbial diversification, at least with respect to communities inside aggregates. However, as aggregates also sequester C, calcium can limit C availability in the surrounding soil, which may adversely impact the overall microbial diversity at depth. Magnesium promotes soil aggregation and hinders root growth, and for these reasons it will have similar implications as calcium for the surrounding microbial community. In addition, magnesium and K have been shown to repress growth of mineral-weathering bacteria (e.g., *Burkholderia*, *Collimonas*) in forest soils, as well as decreasing weathering potential [45]. This may be linked to the fact that weathering is less favorable under conditions when microbes have easy access to minerals, as under such conditions mineral-weathering bacteria lose their selective advantage and may be outcompeted by fast-growing microbes [45]. As a consequence, higher levels of certain nutrients may decrease the rates of soil formation.

Conversely, microbes themselves contribute to soil aggregation and mineral weathering, through the production of high-molecular-weight sugar-based polymers (referred to as extracellular polysaccharides, or EPS). EPS are produced by microbes for a number of reasons, including protection from desiccation and nutrient stresses, as well as adhering to surfaces [46]. Additionally, EPS stabilize soils by gluing soil particles together in aggregates, and increase soil C residence time [46,47]. As for mineral weathering, EPS affect these processes by (1) retaining water around minerals, thereby increasing the potential for mineral hydrolysis; (2) acting synergistically with organic acids to complex with mineral ions and promote dissolution; and (3) using minerals on rock surfaces for biomineralization, sequestering them in organic tissue [48,49,50,51]. Soil depth plays a role in these trends: EPS content and microbial gene abundances for EPS synthesis have been measured to be higher in surface soils than in subsoils (although if the soil system is subject to anthropogenic disturbances such as tillage, the opposite may be true) [46,47,52,53,54]. These trends are likely because EPS synthesis is favored under soil conditions such as high organic matter content, high C/N ratios, and greater root biomass [46,47,52], which are associated with surface soils. However, further work is needed to better elucidate the direct connections between soil depth, microbial EPS production, and soil aggregation/mineral weathering.

Interestingly, the relative strength of individual edaphic variables in influencing communities may also vary with depth. When comparing the vertical profiles of soils under legume cultivation with uncultivated control soils, the strongest influence on microbial community structures in the 0–40 cm layer was K availability, while at 40–120 cm the strongest determinants were pH and N, and at 120–300 cm they were pH and organic matter content [31]. Such differences are likely due to the higher concentrations of nutrients such as K in surface soils [31], allowing them to exert more of an influence at this depth. Meanwhile, nutrients in subsoils are scarcer and often bound in recalcitrant soil organic matter (SOM). Together, these results underline the importance of considering variability in nutrient profiles with depth when studying soil microbial ecology.

### 2.3. Soil Physical Structure

Soil physical characteristics that change by depth or soil horizon include bulk density, water content, texture (i.e., clay/silt/loam proportions), and porosity [10,11,14,27,29]. Compaction and clay content generally increase with depth [55], so subsoils tend to have higher bulk density and lower porosity than surface soils [12,56,57]. As a result, root growth and gas diffusivity are hindered, thereby favoring anaerobic and/or non-root-associated microbes [30]. With respect to porosity, as pores become smaller and less frequent in deeper soils, movement is impeded—not only that of microbes, but also for substrates, nutrients, and oxygen [58,59]. In deeper soils, inputs of moisture and substrates can occur through transport down paths created through bioturbation (e.g., soil cracks created by root growth or earthworm tunneling). The result is that microbial activity in deeper soils will be less homogenous overall and instead more localized to these regions of fresh inputs [60]. Conversely, surface soils have greater porosity and enjoy relatively frequent dry-wet cycles [61,62], which leads to fresh inputs of substrates and nutrients for microbes, and microbial activity can therefore remain comparatively high. Pores also provide protection from predators [63], so microbial survival may be adversely affected in deeper, less porous soils. Soil texture also impacts microbial activity and diversity [64]: clay-heavy soils display greater rates of C mineralization than silty or loamy soils [65], as their higher water-holding capacities allow them to maintain higher nutrient concentrations and support a more compositionally and functionally diverse microbiome [65,66]. As the percentage of clay content generally increases with soil depth [14] (e.g., in a transition from a loamy surface soil to a clay-heavy subsoil) the microbial community composition may be altered accordingly to one more optimized for C mineralization. Bacterial richness and diversity have been shown to positively correlate with clay content [67], both for the aforementioned nutrient capacities as well as protection from predation [68]. However, the relative strengths of higher clay content vs. lower porosity and oxygen concentrations, and how this interplay influences microbial diversity in deeper soils, has yet to be comprehensively investigated.

Farming-related disturbances affect soil physical structure and the microbiome in a manner depending on management intensity [31]. Moderate tillage introduces oxygen and fresh C into deeper layers, thereby stimulating subsoil soil microbial activity and SOM decomposition [69]. However, severe tillage ultimately increases soil compaction and thereby prevents the transport of water, gases, or dissolved C into deeper soil layers [7,56]. A comparison of tillage methods found conventional tillage homogenized C and N contents down the vertical profile, which lowered variability in bacterial community composition between depths, whereas reduced tillage increased variation in C and N and yielded higher heterogeneity in fungal populations [70]. Similarly, when comparing a plowed pine plantation with two undisturbed sites, the plowed site had significantly different trends for microbial taxonomic structures and metabolic capacities compared to the undisturbed sites. For example, functional diversity increased with depth in the plowed site, but the opposite trend was found for the undisturbed sites [71]. Yet another study found a decline in overall microbial activity with soil depth that was most pronounced in an undisturbed grassland and least pronounced in an arable field [17]. These results suggest a homogenizing effect across depths on the soil microbiome exists in agricultural soils. Furthermore, functional trends in soil microbial ecology need to be taken with the caveat that site history and management strategy play a major role. More analyses of the interactive effects of land management and soil depth will help to elucidate the effects of this relationship on the microbiome.

## 3. Changes in Microbial Community Composition with Depth

### 3.1. Changes in Biomass and Diversity

Microbial biomass is substantially lower in deeper soils compared to surface soils [15,24,72], by as much as two orders of magnitude [5], and microbial turnover is also significantly slower [24,41]. The decline in biomass is consistent regardless of the metric used for biomass quantification (e.g., PLFAs, substrate-induced respiration, and chloroform fumigation extraction) [5]. However, site history can affect this trend. One study found, while in natural ecosystems (forests, grasslands) microbial biomass decreased with depth, in an arable field there was no significant change [17]. This was likely a consequence of the homogenizing influence of agricultural activities, as discussed earlier. Conversely, in soil environments that are largely inhospitable at the surface layer (such as biocrusts), biomass levels have been shown to increase with depth, as subsoil communities are shielded from surface stresses such as radiation or desiccation [73]. Regardless, the subsoil represents a significant reservoir of microbial biomass—estimates for the proportion of overall soil microbial biomass contained in subsoils range from ~30% [5,17] to 58% [74] or more, depending on the subsoil depth cutoff used in a given study.

Microbial community diversity indices generally decrease with depth, including Shannon’s diversity [14,15,19], evenness [75,76], and richness [5,28]. Multiple factors are proposed to contribute to this trend, including alterations in soil physiochemical parameters [30,44], C supply [18,30,62], water content [77,78], and an increase in anaerobic conditions [79,80]. With respect to moisture content, over an aridity gradient in Mongolian grasslands, the magnitude and directionality of depth patterns for bacterial and archaeal diversity were highly dependent on where the soil samples fell along the aridity gradient [41]. The differences in bacterial diversity metrics between top- and subsoils were less pronounced with higher aridity, while archaeal diversity showed the opposite trend. Land-use type also affects the nature of the diversity–depth relationship. By examining soils at 10 cm intervals from 0–1 m, from 20 separate sites across the United States, it was found that Shannon’s diversity consistently decreased with depth for both bacterial and archaeal communities, but rates of this decline varied by site [14]. While there were some commonalities across disparate locations (such as negative correlations of total N and soil organic carbon (SOC) with depth) [14], they concluded that the differing rates were likely due to site-specific discrepancies in soil parameters such as sand content or extractable iron. For this reason, it is advisable to measure abiotic soil traits to better understand factors that may influence the diversity–depth relationship.

### 3.2. Changes in Bacterial Composition

#### 3.2.1. Changes in Gram-Positive and Gram-Negative Bacteria

In addition to diversity metrics, changes in the soil microbiome over depth are reflected in the abundance patterns of broad groups of microbes, with one example being the ratio of Gram-positive to Gram-negative bacteria. While actinomycetes and other Gram-positive bacteria have been shown to be common in deeper soils [15], Gram-negative bacteria typically decline with depth [5,13,17,81,82]. These trends are likely linked to nutritional preferences. Many Gram-positive bacteria are oligotrophic and predominate in nutrient-poor environments such as subsoils [83]. Meanwhile, the more copiotrophic Gram-negative bacteria are often dependent on labile C, which is abundant in surface soils, provided by plant litter and other similar sources [83]. Thus, the Gram-positive to Gram-negative ratio has been shown to mirror C availability by soil layer [84], especially as both groups appear to maintain consistent substrate preferences regardless of which soil layer they are located in [84]. In addition to C preferences, some Gram-positive bacteria form endospores to tolerate adverse conditions such as desiccation, anoxic, or low-nutrient stresses [85], which helps to explain their persistence in subsoils [2,86,87]. That being said, a comparison across many soil types found that two Gram-negative phyla (Chloroflexi and Nitrospirae) were consistently associated with deeper layers [14]. In other studies, Gram-negative phyla (e.g., Acidobacteria and Bacteroidetes) or Gram-positive (Actinobacteria) have not been seen to vary significantly with depth [19,28]. While this may be attributable to similar C contents over depth in these specific soils, it is also likely there are other characteristics than Gram-staining classification or substrate preference that contribute to bacterial survival in different soil layers, as outlined below.

#### 3.2.2. Changes in Bacterial Phyla

Depth-related shifts in the microbiome can also be seen at the phylum level (reviewed in Figure 1), likely due to the shared characteristics of phylum members causing them to respond to depth in a similar way. Across a diverse array of soil depth studies, the bacterial phyla most commonly enriched in surface soils include Cyanobacteria, Planctomycetes, and Proteobacteria [16,19,28,31,76,88,89,90,91,92], while for subsoils this list includes Chloroflexi, Firmicutes, Nitrospirae, candidate phylum GAL15, and Gemmatimonadetes [14,16,19,76,88,89,92]. Many other major phyla show contrasting trends in different studies, possibly due to class-, genus- or species-level discrepancies within a given phylum, or alternatively due to differences in soil/environmental characteristics or even sampling protocols between studies. Regardless, a number of reasons have being posited as contributors for observed phylum-level enrichment trends, as discussed below.

The physiological and metabolic traits of microbial taxa may help to explain their depth abundance trends. Certain representative members of the dominant phyla in deeper soils (including Actinobacteria [92], Chloroflexi [15,28], Firmicutes [16], and Verrucomicrobia [15]) are capable of using more recalcitrant C sources and/or are tolerant of low-nutrient conditions. By contrast, many members of less stress-tolerant phyla (Proteobacteria [30,44], Arthrobacter [93], and Bacteroidetes [15,30]) are often depleted in these layers. Factors other than C or nutrient availability will also influence abundance patterns. For example, some members of Gemmatimonadetes are adapted to low-moisture conditions [94,95], and were found to be enriched in drier subsoils [28]. Similarly, Verrucomicrobia have been found to be prevalent in intermediate soil layers (e.g., 10–20 cm [30] or 20–40 cm [15]), possibly due to their preference for microaerobic rather than fully aerobic or anaerobic environments [30]. Conversely, Nitrospirae have been shown to increase in abundance with depth [14,90,93], which is attributed to the nature of their interactions with other microbes [28,89]. The growth of Nitrospirae can be repressed by heterotrophic, fast-growing microbes in the C-rich area around plant roots. However, in deeper soils Nitrospirae growth is less obstructed as root biomass levels go down [89]. Alternatively, Nitrospirae generates energy through ammonia or nitrite oxidization (the only known bacterial group with both pathways [96]), which presents a selective advantage for survival at depth. By contrast, light-dependent microbes and related genes are rare where sunlight is limited [97], which could explain why photoautotrophic Cyanobacteria often decline in deeper soils [16,88,98]. Similarly, the lower abundance of Proteobacteria in biocrust subsoils was attributed to a decline in family Rhodobiaceae, which is photoheterotrophic [88].

Some phyla show contrasting depth patterns depending on the study. For instance, Actinobacteria have been measured as increasing in abundance with depth [5,31], decreasing [76,89], showing no significant changes [28], peaking in mid-level layers [91], or being at their lowest levels in mid-level layers [15]. Given the vast metabolic diversity of Actinobacteria, this may be unsurprising as the different members may fill disparate soil niches. Actinobacteria comprise a major part of the microbiome in environments as diverse as alkaline soils, SOC-rich soils, low-nutrient soils, and anoxic soils, and additionally can form spores to enter dormancy if soil conditions become too harsh [99,100]. Therefore, Actinobacterial abundances may be dependent on the site characteristics (e.g., if deeper soils happen to be more alkaline, you may see more Actinobacteria at depth). It is possible that differential relative abundances of Actinobacterial subgroups bias overall Actinobacterial abundances by soil layer. Contrasting depth trends are often seen between the individual subgroups of other bacterial phyla, including Verrucomicrobia, Chloroflexi, Acidobacteria, and Proteobacteria [44,76,89], which could hold true for Actinobacteria as well. For this reason, it is worth considering the community depth trends below the phylum level.

#### 3.2.3. Changes in Bacterial Classes

Depth trends are by no means universal across all members of a phylum. For instance, while the Proteobacteria phylum typically decreases in abundance with depth, its constituent classes either generally mirror this trend (i.e., Alphaproteobacteria [15,19,89]) or show different patterns between studies (e.g., Betaproteobacteria and Gammaproteobacteria [15,16,19,72,89]). The metabolic predispositions of subgroups are a likely explanation. For example, in one soil profile, Alphaproteobacteria were more abundant in surface soils; however, Deltaproteobacteria were enriched in deeper soils [44], which was attributed to its constituent genus Geobacter’s ability to use iron oxides during respiration in anaerobic subsoils [44]. Another striking example is the phylum Acidobacteria, whose subgroups have been repeatedly observed to have contrasting abundance patterns by soil layer [44,89]. In soil aggregate communities at multiple soil depths, Acidobacteria subdivisions 1 and 2 were associated with depths that were low in nutrients, saturated in water (i.e., anoxic), and acidic [44]. Meanwhile, soil depths with higher nutrient contents were enriched for Acidobacteria subdivisions 3, 4, and 6 [44]. Some Acidobacteria are acidophilic [101], and therefore less prevalent in soils where pH increases with depth [92,98], and more prevalent in subsoils where pH follows the opposite trend [16]. Yet, Acidobacteria also includes aerophilic [102] and oligotrophic [83] groups, traits that would, respectively, favor their enrichment in surface and deep subsoil layers, complicating matters. If metabolic proclivities vary substantially between these Acidobacterial subgroups, this could explain why some subgroups exhibit different depth trends. Likewise, Proteobacteria abundance correlates with pH (although both positive [90] and negative correlations [31] have been measured). Again, contrasting abundance trends between studies may be dependent on which Proteobacterial class is dominant and the optimum pH of its constituent members [103,104].

### 3.3. Changes in Archaeal Phyla

At a broader level, as soil conditions shift with depth, there are changes in the abundance ratios between archaea, bacteria, and fungi, due to their distinct metabolic dispositions. Notably, archaeal communities often display contrasting patterns to bacteria or fungi. The archaea:bacteria ratio generally increases with depth [105,106,107], and the diversity of archaeal communities increases, while bacterial and fungal diversity decrease [22,108]. As archaea are linked to anaerobic, low-nutrient, or other extreme environments [109], their enrichment reflects a propensity for survival in the less resource-rich subsoils [110,111]. For example, an analysis of a New Zealand soil chronosequence found soil mineralogical profiles that were increasingly nutrient poor as depth linked to higher archaea:bacteria ratios [107]. Indeed, one of the most enriched archaeal groups was the Bathyarchaeota phylum [107], which is characterized as having members capable of surviving in nutrient- and energy-limited environments due to its metabolic plasticity [112].

The major archaeal phyla in soils are Euryarchaeota, Crenarchaeota, and Thaumarchaeota. Euryarchaeota are generally found in deeper soil layers [30]; in fact, in one analysis incorporating multiple soil sites, it was the only archaeal group to be consistently located in subsoils [14]. This association could be from the ability of some of its members to oxidize methane under anaerobic conditions characteristic of subsoils [113]. That being said, another study found Euryarchaeota DNA sequences in surface soil layers only, suggesting that some members of this phylum can in fact tolerate aerobic conditions [114] and thus oxygen sensitivity may not be the best explanation for their abundance patterns. Crenarchaeota and Thaumarchaeota have also been found in deeper layers [114,115], likely as both contain anaerobic ammonia oxidizers and denitrifiers [44,116], but subgroups within these phyla show differing abundance trends with depth [44,117]. As with the previously discussed bacterial phyla, there are distinct ammonia oxidation capacities or other metabolic traits between subgroups that may allow them to be differentially abundant between layers.

Local site characteristics can also influence changes in archaeal abundances. When analyzing archaeal communities across nine soil pits in a forested montane watershed in Colorado, some pits showed an increase in archaeal abundance with depth while others were static [15]. The authors assumed this related to ammonia content, which was higher in some pits than in others, and in turn resulting in ammonia-oxidizing archaea such as Thaumarchaeota being enriched in deeper pit layers in a manner linked to ammonia abundance [115]. In addition, soil tillage may disproportionately harm archaea relative to bacteria or fungi. Many archaea have comparatively slow growth rates [118] and thus lower resilience to soil disturbances compared to bacteria [119,120]. Furthermore, a large proportion of archaea are obligate or facultative anaerobes [121], and are sensitive to high-O_2_ conditions. Therefore, farming practices that result in aeration of deeper soil layers will adversely affect subsoil archaea [22].

### 3.4. Changes in Fungal Phyla

In contrast to archaea, fungi are often more enriched in surface soils than subsoils. Fungal:bacterial ratios decrease with depth [8,14], as well as fungal biomass, diversity, and ITS sequence copy numbers [122,123]. While total microbial biomass generally decreases with depth, fungal biomass tends to decline more steeply than does bacterial biomass [27,124]. Similarly, the proportion of active fungi vs. total fungi has been demonstrated to decline faster than the same proportion for bacteria [27]. The dearth of fungi in deeper soils can be tied to oxygen levels and substrate availability. Firstly, fungi are largely aerobic and their preferred substrates are plant polysaccharides [125,126]. Observed declines in fungal:bacterial ratios with depth have been attributed to lower amounts of labile plant litter, compounded by anoxic conditions [107]. While some fungi have the enzymatic capacity to degrade the recalcitrant SOM that is found in deeper soils, they require an initial contribution of labile C in order to do so, which is often in short supply [124]. Another contributing factor is soil chemistry: bacteria are more common in alkaline soils than fungi [127,128], so fungal:bacterial ratios will decrease from surface to subsoil layers in soils where pH goes up with depth [15,19,28]. In addition to pH, fungal:bacterial ratios are significantly positively correlated with C/N ratios [129], possibly due to different stoichiometric optima between bacteria and fungi (bacteria generally require more N than fungi) [130]. Therefore, when soil C/N ratios go down with depth [15,19], fungal:bacterial ratios will be expected to decrease as well.

Multiple studies have observed little to no significant change in fungal community composition over depth [91,122,131]. This could be due to the low fungal biomass in deeper layers that makes it hard to discern any consistent trends. Alternatively, the ability of multicellular fungi to form hyphae and thus expand their reach across spatial scales permits them to persist at multiple depth profiles simultaneously [131]. This means that fewer fungal taxa will be significantly associated either with only surface soils or with only deeper soils. Very often, the only visible taxonomic trends are at the genus or operational taxonomic unit (OTU) level, as seen in one study where only 8 fungal OTUs (and no higher-level groups such as genera or families) significantly increased with depth [122]. In another case, the only change was an increase in the *Inocybe* fungal genus (up to approximately 46% of all fungal read counts) in deeper layers, likely as *Inocybe* is a plant symbiont and thus enriched in the presence of roots [132]. At the phylum level, soils are dominated by a few fungal phyla (Ascomycota, Basidiomycota, occasionally Zygomycota). Ascomycota generally decreases with depth [70,90,108,133], as its members are primarily saprotrophic and therefore localize at surfaces where plant litter is found. Zygomycota may increase with depth [70,108,132], while Basidiomycota tend to decrease [90,108,132]. Correlations with soil factors such as P or iron content contribute to these trends [108], but inconsistencies between studies as well as the low fungal levels in subsoils both prevent easy explanation of fungal community shifts.

### 3.5. Plant Presence and Soil Microbiome Depth Trends

The presence and types of plant roots in a soil ecosystem substantially influence soil chemistry, structure, aggregation, and nutrient content. For this reason, such soil factors differ between vegetated soils and bare soils [6,134], or between fields of different land-use types [135,136,137], and these differences include how they change over depth. For example, CO_2_ concentrations were shown to increase with depth to a far greater extent in planted than unplanted soils, due to root respiration [134]. Planted soils also have different microbiota and gene expression patterns [138,139,140]. There are often higher ratios of copiotrophic to oligotrophic bacteria [141] in planted soils due to the C substrates and detritus originating from root turnover. Unfortunately, depth patterns have not been extensively studied with respect to the planted-unplanted soil division. Some studies have demonstrated that soil microbial biomass declines more sharply with depth in planted soils, again due to the higher concentration of organic material at the surface of these soils, although the overall amount of biomass is comparable between planted and unplanted soils at the lower depths [82,134]. For this reason, microbial community differences between planted and unplanted soil microbiomes may only be significant for surface soil layers and not deeper ones [82]. More work will be required to confirm what differences exist in soil microbiome depth trends in bare vs. vegetated soils.

Better addressed are how root-associated (“rhizosphere”) microbial communities change by depth. Observations of rhizosphere communities differing over the length of the root [142] are attributed to a number of factors, including root zone-specific differences in gas concentrations, pH, or release of volatile chemicals [143,144,145]. More directly, plants actively recruit microbes through rhizodeposition of carbon- and nutrient-containing exudates, the makeup of which can be tailored to foster specific communities. Young roots have greater exudation rates in general [93], and release different compounds compared to mature or senescent roots [142,146]. For instance, young roots secrete sugars or organic acids to attract a broad range of microbes, while mature and senescent roots release more recalcitrant C compounds and/or secondary metabolites, enriching for more specialized communities [146,147]. Given that young roots are typically found at deeper soil layers than mature roots, there likely exists a connection between root depth, root age, exudate release and microbial recruitment patterns. Indeed, depth has repeatedly been found to be a significant variable driving rhizosphere microbiome composition [93,143,145,148,149]. However, the relationship between the rhizosphere microbiome and depth is not necessarily linearly correlated. For example, in deep soils a diverse rhizosphere may be attributed to the high exudation levels from actively growing root tips, but at the same time, surface-level rhizosphere communities can be similarly rich due to high SOM content at the surface. Meanwhile, mid-level rhizospheres may be comparatively less diverse due to a lower C content [93,150]. In addition, deeper rhizospheres tend to select for a more consistent community composition between replicates [148,149], as they are less subject to fluctuating environmental conditions or diurnal cycles.

Rhizosphere depth trends are not universal across plants—they are closely tied to the plant in question and its root architecture (mass, length, morphology) [144,151]. In one comparative study, ryegrass (whose roots are largely branched and fibrous) had spatially homogeneous rhizosphere communities. Meanwhile, alfalfa’s taproot system induced a strong depth gradient in rhizosphere chemistry and C content, leading to differing levels in taxa including Actinobacteria and Gemmatimonadetes [145]. Abundance patterns can also be tied to the rhizosphere functional profile. For example, a study of wheat roots found that Firmicutes were the predominant C cyclers in the subsoil rhizosphere, while Proteobacteria were favored in the topsoil rhizosphere [93]. As for rhizosphere fungal communities, there have also been reports of significant variations by depth [87,152], associated with changes in rhizosphere chemistry. For example, rhizosphere pH will foster different levels of bacteria and fungi across depths, as fungi are normally more favored in zones of the rhizosphere with higher acidity [87,153]. Similar to bacteria, fungal communities in the rhizosphere do not necessarily decrease in diversity with root depth; there are many fungal taxa elevated or exclusively found around roots in deeper horizons [152,154]. The chemical nature of soil layers may bias fungal survival: many mycorrhizal fungi associated with deeper roots are better adapted to live in lower, more mineral soils, while those around shallower roots are optimized for the surface organic layer [152].

### 3.6. Changes in Microbial Network Patterns

Interaction networks are used to represent patterns of microbial co-abundance, thereby informing ecological theories about cooperation and competition among soil community members. The strength of microbial interactions has been shown to increase in deeper soils [22], which may be linked to the scarcity of nutrients, and thus a greater degree of microbe-microbe competition or synergy required to acquire them. Investigation of fungal communities by soil depth in a no-till wheat field revealed substantially different network patterns by depth [133]. Networks at deeper layers had fewer significant co-associations but became denser, indicating that the microbes that remain at depth were more likely to have biologically relevant interactions [133]. Another study in native and exotic grasslands found similar patterns [155]. While these interactions may be competitive, another explanation for stronger associations at depth may be a higher degree of syntrophic, cooperative interactions that are required to break down complex SOM. However, given that networks from deeper soils show higher relative abundances of negative interactions than those in surface soils [16], competitive interactions may prove more likely. Network structures have also been shown to differ with depth. In one study, surface microbial communities displayed high network modularity, while subsoils had higher connectivity [16], indicating that networks in deeper soils are more likely to be destabilized by the removal of key members. These results reinforce the importance of interactions between microbes in deeper soil layers.

## 4. Changes in Microbial Activity with Depth

Across multiple land-use types, patterns and overall levels of microbial activity consistently vary with depth [26,75]. Soil enzymatic activity can go down [17,156] as high as 300-fold [90] when comparing surface soils to deeper soil layers. At depth, activity is likely restrained by lower nutrient concentrations, temperature differences [1], reduced C quantity and lability [13,17,135,156], physical separation of microbes from degradable substrates and nutrients [4,157], and anaerobic conditions [10]. For example, in clay soils cropped with reed canary grass, normalized activity was half as high in soils taken from a 30–55 cm depth than those from 0–25 cm [1]. This difference in activity was hypothesized as being due to poorer substrate quality and decreased oxygenation in deeper layers. Microbial activity in deeper soil layers is largely localized to irregular “hotspots” of fresh C inputs, such as roots [4]. As a result, microbial activity and population distribution are both more limited, and more spatially heterogenous in deeper soils [1,158]. By contrast, at the surface, a wider distribution of C and nutrients supports higher, more uniform levels of microbial activity.

Moisture, or lack thereof, also influences activity [159]. Soil wetting events release sequestered C from aggregates [160], promote C and nutrient diffusion, and support microbial motility. Wetting events affect deeper soils less directly than they do the surface [40], but can still induce a measurable increase in subsoil microbial activity. Temperature also influences microbial activity, elevating process rates and shortening doubling times [161]. Surface soils are on average hotter and experience greater temperature extremes than deeper soils [162], meaning that microbial metabolism and enzymatic activities are stimulated by temperature to a greater extent in surface layers [161,163] unless temperatures reach inhibitory levels. Meanwhile, the lower temperatures in subsoils generally present a constraint upon microbial respiration [4].

It should be noted that lower overall activity may simply be a consequence of reduced microbial biomass, and that normalized levels (e.g., normalizing activity against total microbial biomass carbon) in subsoils can be comparable to or even higher than those for surface soils [17,18,156]. For example, C mineralization may occur at similar rates [2] or even more rapidly for subsurface microbes upon substrate addition [3,81,164], which is likely related to how chronically C-starved they are. At six soil depths on a riparian hillslope, normalized activity rates were higher for deeper soils than surface soils, and peaked around 50–100 cm [156]. Despite normalized activity remaining at similar levels, microbial metabolic diversity has been shown to decline with depth in natural ecosystems [7,71], as harsher conditions restrict the types of microbes that can survive under these conditions. However, as with taxonomic diversity, in sites subject to anthropogenic influences such as cultivation, functional diversity has been shown to increase with depth due to the adverse effects of farming practices on surface communities [71].

### 4.1. Changes in Functional Guilds

#### 4.1.1. Anaerobic Guilds

One common trend in deeper soils is higher prevalence of microbes that do not require oxygen. Lower porosity and higher saturation in deep soils decrease gas concentrations per soil volume [10] with the threshold below which oxygen is largely absent varying by soil type [165]. Respiration by soil microbes and plant roots elevates the proportion of CO_2_ to O_2_ [166]. In addition, the lower diffusivity of clay in subsoil impedes efficient exchange of respired CO_2_ with the atmosphere [27], thereby retaining it at this level. For these reasons, anaerobic conditions increase with depth, impacting the microbiome. For example, in subsoils there are soil aggregates with anoxic “microsites” [167], which house anaerobic communities capable of utilizing a diverse array of redox acceptors (NO_3_^−^, SO_4_^2−^, etc.). Conversely, microbial groups that are aerophilic and sensitive to oxygen-poor conditions (e.g., some members of the Acidobacteria) are often absent or rare in deeper soils [92]. Relative activities of anaerobic respiratory processes (e.g., methanogenesis, chemolithotrophy) tend to be elevated at depth [10], and the guilds responsible for anaerobic processes (anaerobic methane oxidization [168], methanogenesis [169], iron reduction [170,171] and sulfate reduction [13]) often increase in abundance. Methane-oxidizers such as Euryarchaeota (anaerobic [172]) or Verrucomicrobia (micro-aerobic [15]) peak in abundance in subsurface layers [30], particularly at the oxic-anoxic interface as some methane-oxidizing bacteria are facultatively anaerobic [173,174]. Other subsoil-associated anaerobic phyla include Chloroflexi [14,88] (which contains anaerobic classes such as Anaerolineae [175]) or Firmicutes [44,98] (contains class Clostridia, which uses fermentative metabolism [175]). Although members of the Actinobacteria are predominantly aerobic [99], observations of Actinobacterial enrichment at 81 cm depth of an Arctic peat soil [98] was proposed to be due to the capacity of certain Actinobacteria for anaerobic respiration of soil organic C through syntrophic fermentation [100]. While the overall abundance of an anaerobic guild can increase with depth, the taxonomic makeup within that guild may be altered. For instance, analysis of iron-reducing communities over soil depths in a Tennessee watershed showed that the proportions of classes varied across depths, e.g., with iron reducers within Deltaproteobacteria being most prevalent at the surface, and those within Gammaproteobacteria were prevalent at the deepest layer [44]. These findings likely reflect other metabolic characteristics, unrelated to iron reduction, of the various guild members that bias their survival across the vertical profile.

#### 4.1.2. Starvation Responses

Microbes adapted to deeper soils often have altered genetic profiles to deal with the stress of this environment. Genes for central metabolism have been observed to increase in relative abundance, while those for secondary metabolism decrease [72]. These differences reflect the need to focus on pure survival in deeper soil layers by emphasizing their core metabolism, rather than apportioning C and energy towards auxiliary processes. The ability to withstand starvation conditions represents a selective advantage to microbes in deeper soils. In one study analyzing genomes from deeper soils, the dominant bacterium at deeper levels had a genome that was enriched for starvation and stress response genes, such as spore formation or using carbon monoxide as a carbon/energy source [14]. As microbial turnover is normally also lower in deep soil, cell growth and membrane formation are slower, with corresponding reductions in fatty acid synthesis [72]. Survival can also come through approaches relating to quiescence. For instance, the relatively high abundance of the Firmicutes phylum in deep soil was linked to endospore formation [150], which would help the cells to endure until substrates were more widely available.

## 5. Microbial Nutrient Cycling and Soil Depth

C, N, and P are essential components of microbial biomass, and their availability (or lack thereof) impacts both the soil microbial community composition and biomass levels [19,176,177]. Soil C, N, and P generally all decrease with depth [14,15,19,135,178,179], as fresh inputs from surface litter and other plant debris become increasingly limited [180,181]. For example, due to the decreased presence of roots and litterfall at depth [2], C levels were shown to be lower in subsoils [4,182], in one case 478 times lower at 60–80 cm compared to 0–20 cm [183]. Even when it becomes available, labile C is rapidly mineralized by subsurface microbes [81,164]. As such the ratio of C to N usually decreases substantially with depth [15,90,136]. Changes in nutrient ratios has repercussions for soil microbes, as they are a significant determinant on soil microbial community composition [5,28,29,176,177]. At the ideal stoichiometry of C, N, and P (the “Redfield” ratio, approximately 60:7:1 in soil [184]), microbial growth and anabolism are stimulated. Deviations from this ratio lead to one or more nutrients becoming too low for optimal growth [184,185]. Furthermore, even if stoichiometry were to be ideal, subsoil sources of N or P are often in chemically inaccessible forms [186] (such as mineral N being adsorbed onto clay surfaces [180]) and microbial growth can still be hindered.

### 5.1. Carbon Cycling

#### 5.1.1. Carbon Availability by Depth

Globally, of the approximately 2344 gigatons of C in soils up to 3 m, only about 615 are found in the top 0.2 m [135,183], indicating there exists a substantial amount of SOC located below the surface. Processes that contribute to subsoil organic C include rhizodeposition, the decomposition of dead plant tissue, and a downward translocation of dissolved organic matter [4,133,180,187]. Which process dominates in a given environment depends on the soil type and plant cover [188]. Plants are more likely to deposit C at greater depths in Vertisols (clay-heavy soils) than other soil types, due to the predisposition of this soil type to crack under swelling-shrinking cycles and allow substrates to travel deeper [188]. External factors such as land use, climate, or topography, or environmental disturbances [166,189] also play a role in C distribution with depth. For instance, erosion causes redistribution of significant amounts of C downhill, from shoulder slope to foot slope positions. As for land use, significant quantities of C will be respired and CO_2_ lost to the atmosphere if soils are subject to physical disturbances such as tillage [166].

Stabilization mechanisms play a key role in C availability, especially in deeper soils. The three principal stabilization mechanisms include: (1) physical protection through aggregate formation, (2) formation of physicochemical interactions (e.g., through organo-mineral associations with iron or aluminum oxides [190,191]), and (3) chemical recalcitrance to degradation [180]. With respect to the first two mechanisms, many subsoil C compounds have charged chemical properties [13], causing them to be integrated into soil aggregates or adsorbed onto minerals [156], and as a result the proportion of mineral-bound SOC generally increases with depth [183]. Interestingly, while soil aggregates are found across soil depths, they play a greater role in C stability in subsoils. In one study, after sieving a diverse array of soil samples to disrupt aggregates, subsoils exhibited significant increases in C mineralization whereas surface soils were comparatively unaffected [7]. The higher level of stabilization in deeper soils is likely at least partially due to microbial necromass, which is more prevalent at deeper soils than surface layers [192]; the C inside microbial necromass is primarily stabilized via the first two mechanisms [193]. As for chemical recalcitrance, subsoil SOC often contains compounds such as lignin that require specialized activities for their breakdown and are thus less amenable to microbial metabolism [180]. However, as degradation of subsoil complex C readily occurs upon supply of labile substrate [194], it has been argued that a compound’s inherent recalcitrance is not as limiting a factor so much as lack of labile C to prime microbial degradation thereof [195]. The importance of these three mechanisms for C stability in deeper soil layers is hotly debated, especially as their relative contributions in a given soil can be influenced by characteristics such as pH or mineralogical composition [196]. For instance, if pH decreases with depth, higher acidity can promote complexation of iron or aluminum ions with organic matter in these subsoils, thus favoring mineral associations as the primary method to stabilize C [197].

#### 5.1.2. Carbon Kinetics by Depth

SOC decomposition kinetics change with soil depth for several reasons, including microbial functional capacity, substrate availability, and C lability. Firstly, subsoil microbes have less collective functional diversity with regard to C metabolism than do surface microbes [7]; the majority of subsoil C exists in relatively stable forms [156], so it makes more biological sense for a subsoil microbe to invest energy towards breakdown of one recalcitrant substrate that is readily available, rather than several labile but scarce substrates. Along the same lines, another issue is substrate supply. Overall rates of C mineralization are demonstrated to decrease with depth [198], primarily limited by lower C availability [2,183,199,200]. Indeed, relative to surface soil microbes, lipid profiles for subsoil microbes had higher ratios of cyclopropyl:monoenoic precursors and total saturated:monounsaturated fatty acids, indicators of C limitation [201]. This problem is compounded by limited diffusion: available substrates and microbial biomass decrease in concentration to the point where microbe-substrate connections become much more difficult to establish and are increasingly reliant on diffusion to do so [157,202]—but the lower diffusivity in deeper, more compacted soils (as discussed earlier) limits these connections.

However, normalized C mineralization rates for subsoil microbes can be as high as those for surface microbes if fresh C is provided [7,203] (which rarely occurs outside the context of laboratory incubation studies), or if the microbes can overcome the greater “activation threshold” of time and energy required to initiate mineralization in deeper soils. A long-term incubation of litter bags at different soil depths found that after 6 months, less C was mineralized by microbes in subsoil than in surface soil litter bags. However, after 3 years C concentrations were comparable between depths [203]. In a similar work, root material was added to soils at different layers (20, 60, and 300 cm) and C mineralization rates measured. There was a ~5-day lag period for microbes at lower soil levels compared to the microbes in surface soils, possibly as the nutrient-poor conditions at depth prevented microbes from reaching the activation thresholds as soon as they would in at the surface [204]. Taken together, these results suggest that, while microbes may be slower to respond in deeper soils due to their lower starting biomass and slower growth rates, they can have comparable mineralization rates once they multiply to sufficient numbers [203].

Interestingly, C dynamics display different responses in subsoils than surface soils upon environmental disturbance. It would be expected that subsoils would be reasonably buffered against perturbations [58] as they generally lie below the “damping depth”, or the depth below which environmental fluctuations do not generally affect the soil profile. However, some studies have actually measured higher responses for microbial activity in subsoils than surface soils as a function of seasonal changes in the environment [18]. Such microbial activity responses may depend on the type of fluctuation affecting the soil. It has been shown that C mineralization in subsoils is more likely than in surface soils to be disrupted by elevated CO_2_ [75] or high temperatures [1,81,186], although it is more resistant to drought stress [186]. For temperature stress, the aforementioned trend could be linked to the types of C substrates present at different layers—recalcitrant C is generally more temperature sensitive (i.e., decomposition kinetics change more readily under fluctuations in temperature) than labile C [205,206]. C mineralization being affected by temperature is also likely related to C concentrations. If subsoil mineral layers are greatly reduced in C, temperature sensitivity will instead be lower there than in SOC-rich surface layers [207]. Alternatively, temperature sensitivity may also be tied to distinct thermal optima between soil microbiomes at different depths, a consequence of evolutionary adaptations to the different temperatures by soil layer [186]. As for drought stress, simulating moderate drought has been shown to affect respiration to a greater extent in surface soils than in subsoils [186]. The chronic fluctuations in moisture availability in surface soils may produce microbial communities better evolved to resist drought stress than subsoil communities.

#### 5.1.3. Changes in Carbon Metabolism Patterns by Depth

Metagenomics approaches [97,208,209,210] have been used to investigate microbial metabolic strategies down the vertical soil profile. For instance, metagenomics has provided evidence for segregation of microbial functions by depth, with surface soils being enriched for pathways for oxygen or light sensing and bacterial chemotaxis, whereas deeper soils were enriched for inorganic N metabolism or various archaeal pathways [97,208,209]. In particular, it has been shown that enrichment for C pathways differs by depth, including patterns for the metabolism of different amino acids [208], or higher CAZyme diversities and levels of inorganic C processing in shallower soils [209,210]. It should be noted, however, that factors such as soil type [97] or even DNA extraction methodology [208,211] may lead to different depth trends between soil metagenomes. Another approach for studying C usage patterns by depth is to use profiling assays [7,75]. Biolog MicroPlates^TM^ have shown that surface soils have higher rates of overall C usage than deeper soils, up to an order of magnitude greater [7]. In a similar study using GeoChips, major processes (carbon fixation, TCA cycle, polysaccharide decomposition, etc.) were detected at both 0–5 cm and 5–15 cm layers, but there were smaller subsets of genes for the same processes at the lower layer [75], pointing to a narrowing of metabolic complexity with depth.

Substrate utilization patterns can also differ with depth. A commonly accepted trend is that surface soil microbes mainly use easily decomposable compounds derived from plants, while subsoil microbes degrade recalcitrant compounds such as decades-old SOC or microbial necromass [84]. However, this is by no means a universal trend—subsoil microbes often use simple substrates and vice versa [29], and the most common C substrates in soils (e.g., carboxylic acids, phenols, amino acids) are consistently metabolized across all depths, though exact usage patterns can vary [71]. This trend could be linked to the availability of plant root exudates, which are more likely to penetrate surface layers than deeper soils [91]. Studies have repeatedly demonstrated that root exudation decreases with soil depth [212,213]—in one study of beech roots, when the already-substantial decrease in exudation per root biomass was coupled with the lower root biomass with depth, root exudation per soil volume went down 98% from 2 cm to 130 cm below the surface [213]. On an individual basis, exudate compounds may have slightly different depth abundance patterns depending on their chemical properties, microbial consumption rates, and their propensity for root reuptake. For example, in one model, negatively charged organic acids had a higher net efflux from deeper roots compared to sugars or amino acids, given that the negative charge of the root and limited influx mechanisms prevents organic acids from being recaptured easily [214]. In addition to substrate availability, the microbes themselves may be more metabolically inclined to use certain types of substrates. When performing community-level physiological profiling on subtropical forest soil layers, surface microbes were found to be more likely to use simple substrates (carboxylic acids, carbohydrates) while subsoil microbes were more likely to use amino acids or amines [29]. In another case, surface soil (5–10 cm) microbes had a greater propensity to use polymers and disaccharides while subsoil (80–100 cm) microbes were enriched for usage of root exudate compounds (e.g., amino and organic acids) [7]. Similarly, when comparing respiration rates between surface soil and subsoil layers of two silt loams, the compounds that provoked the largest differences were sugars and amino acids (alanine, L-asparagine, and L-serine) [124]. Taken together, these results implicate amino acid metabolism as being a differentiating factor between soil layers, possibly as a consequence of N limitation at depth. As for recalcitrant C, contrary to expectations, certain studies have found that oligotrophic C metabolism goes down with depth [72], which could be due to a number of causes. It is possible that the occlusion of C substrates through aggregation or organo-mineral association [7], the lack of labile C to prime SOM breakdown, and/or the slower rates of degradation of complex C, together contribute to microbial functional profiles being dominated by genes and transcripts for usage of fresh C from plant root exudates—exudates being the only significant source of fresh C in this layer.

### 5.2. Nitrogen Cycling

Nitrogen content is generally negatively correlated with soil depth [14,15,19,28,72], which adversely impacts biogeochemical cycling in deeper levels. Firstly, as N is necessary for enzyme synthesis, its limitation will decrease enzymatic process rates, including those related to SOC mineralization [186]. There have also been reports of decreases in N-cycling functional gene abundances [86,179], gene categories [72], and N-cycling microbial biomass [179,215] with depth. In other studies, however, little to no difference was found in overall N decomposition kinetics [203] with depth. Instead, different N-cycling guilds dominated in the various soil layers in response to availability of various N sources. The relative abundances of ammonia oxidizers (both bacterial or archaeal) [44,216], nitrifiers and denitrifiers [215], and diazotrophs [217] have all been seen to be altered with depth. For instance, nitrifiers and diazotrophs are commonly associated with surface organic layers rather than deeper mineral layers [2,86]. With respect to diazotrophs, their prevalence in surface soils is a consequence of higher C/N ratios in organic layers favoring N fixation, as free-living diazotrophs have been shown to prefer C-rich soils [218] including the rhizosphere. In deeper soils where C/N ratios are lower, ammonia oxidation may prevail over other, more carbon-intensive processes, such as diazotrophy or N mineralization. As a result, bioavailable N can be produced without requiring a substantial C investment, and C/N ratios can be maintained at a workable level [179]. Indeed, ammonia-oxidizing microbes were found to be enriched in subsoils in some studies, particularly ammonia-oxidizing archaea [44,219]. The relative abundance of ammonia oxidizers has also been tied to soil pH and depth changes thereof, as ammonia-oxidizing archaea have been shown to be more competitive than their bacterial counterparts in acidic, low-OM conditions [220,221] such as those of subsoils.

Local environmental characteristics such as soil type or N distribution influence how abundances of N-cycling microbes change with depth. When comparing N-cycling functional genes across soil environments, subtropical soils were shown to have significantly different patterns than temperate soils. In temperate soils, diazotrophs were prevalent at all four depths sampled, which was thought to be attributable to N limitation throughout the profile [179]. Yet, for subtropical soils, in which soil N levels suffered steep losses from surface to deeper layers, both diazotrophs and N-mineralizing bacteria decreased in abundance with depth, while the abundance of ammonia-oxidizing microbes increased [179]. Likewise, in an analysis of Chinese rice paddies and converted fields, the distribution of soil nitrate across sites was correlated with abundances of nitrifiers in 0–20 cm surface layers, but with denitrifiers in 80–100 cm subsoil layers [215]. The N balance between soil layers is an important consideration. In an Inceptisol soil (a class of soil that lacks a well-defined profile or substantial horizon development, found in humid and subhumid regions), the greater activity of surface soil diazotrophs was hypothesized to counterbalance the N losses from weathering deeper in the soil profile [86]. Comparisons to an Oxisol (soils that are highly weathered, with low fertility, found in intertropical regions) found no such vertical differentiation [86]. As a result, depth trends for N cycling should be taken in the context of N concentrations, and the strength of the N concentration gradient.

### 5.3. Positive Priming

As discussed, microbial metabolism of complex SOM is stimulated by (and arguably dependent on) inputs of labile C for a quick boost of energy, generally supplied to them by the plants through labile root exudates, or else as substrate transport downward through cracks or bioturbed holes. This stimulatory effect is known as “positive priming” [29,222]. The rarity of such inputs in subsoils suggests that natural occurrences of positive priming would be limited in deeper soils [1]. That being said, under laboratory manipulations, positive priming effects were found to be more pronounced in deeper soils than surface soils [29,183,199,223]. Upon glucose addition to semiarid grassland soils from an array of depths, there was not only more CO_2_ efflux from subsoils, but also a greater stimulation of phenol oxidase and peroxidase [81] (both being extracellular enzymes implicated in polysaccharide degradation [224]). Higher positive priming for subsoil microbes has been hypothesized as necessary for these microbes to produce SOM-degrading enzymes and thereby maintain C and nutrient uptake under limiting conditions [29]. Conversely, priming effects in surface soils are transient, without sustained enzymatic activation or SOM breakdown [225], and quickly return to basal levels [200]. For example, fructose addition to soil layers taken from a wheat-maize chronosequence found that C mineralization peaked early in surface soils with a rapid decline thereafter, whereas in subsoils the peak only occurred after 3 days and declined much more slowly [7], supporting the idea that subsoils exhibit more sustained priming effects. However, the surface–subsoil divide is not as great a factor in determining priming effects as the soil chemical environment: regardless of depth, positive priming effects can be dampened, nonexistent, or even reversed if the soil already contains sufficient C or nutrients [124,199,200]. For instance, adding carboxylic acids to subsoil layers of Luvisol and Cambisol soil types showed more respiration in the Cambisol subsoil, which the researchers believed to be due to Cambisols already containing high levels of C, whereas in Luvisols, the C-limited subsoil microbes did not respire the added carboxylic acids, as the microbes required them for anabolic processes [124].

The elevated positive priming effects in subsoils may also be related to low levels of other nutrients, such as N and P, and the theory of “nutrient mining” [226]. In this theory, in nutrient-limited subsoils, addition of C will prime microbial SOM breakdown, allowing the microbes to “mine” for N, P, or other nutrients by liberating them from SOM [227]. If both N and P are limiting, adding either one in addition to C increases the strength of positive priming relative to the effects of adding C alone [156]. Sometimes no positive priming effects are seen unless N is added alongside C [199]. Evidence of the “mining” theory can be seen for how nutrient addition stimulates various microbial functions. For example, upon N addition in subsoils at 35–65 cm, microbial phosphatase activity increased, indicating that the microbes in this layer pivoted towards P acquisition [200]. By contrast, in surface layers, priming effects of N or P addition were found to be negligible [186] or even reversed [200], as living in the nutrient-rich subsoils could preclude the microbes’ need to participate in SOM breakdown [29].

Digging deeper into this phenomenon, supplying various combinations of C, N, and P across soil layers elicits drastically different trends by depth. In one comprehensive study, while surface soil layers (0–20 cm) had generally little response regardless of the type of substrate addition, upper subsoils (30–100 cm) had greater priming effects under C + N or C + P than C alone. Meanwhile, lower subsoils (>130 cm) had greater effects for C + N + P, though not C + N or C + P, relative to C alone [3]. The authors explained these findings as linked to increasing nutrient limitation with depth. At the surface level, no nutrients were limiting, whereas at the middle level N and P were both somewhat limiting (priming with C + N or C + P stimulated microbes to hunt for the other limiting nutrient) [3]. Concurrently, at the lowest subsoil level both N and P were in such short supply, the soil microbes were only active upon addition of both, as the nutrients were so extremely limiting that mining could no longer occur.

## 6. Conclusions and Future Directions for Modeling Approaches

Here, we have reviewed the tripartite interactions that exist in soils between depth, physicochemical properties, and the resident microbiota. It is becoming increasingly clear that subsoil microbiomes are disparate entities from those at surface layers, with differences in microbial abundance, diversity metrics, community composition, and functional profiles. However, due to a number of factors (for example, lack of understanding of subsoils’ unique attributes and limitations in sampling effort) subsoils have been historically underrepresented. To address this knowledge gap, incorporation of subsoil samples into future soil microbiome studies will deepen our understanding of the spatial variability of the soil.

In particular, a substantial benefit to studying subsoils will be improving the accuracy of computational models for soil functions such as biogeochemical cycling and greenhouse gas emissions. Models tend to rely on surface soil characteristics to extrapolate those of subsoil, despite subsoil itself representing a distinct C source or sink with its own set of parameters [180]. While many parameters relevant to models (e.g., C concentration and lability, root primary production, soil metabolic activity and diversity) are generally highest in surface soils [4], the importance of what remains below the soil surface still remains understated. Models that do explicitly incorporate subsoils tend to assume that subsoils have similar characteristics to those of surface soils, albeit parameterized slightly differently, e.g., reactions occur more slowly [228]. However, newer studies have suggested that, while subsoil decomposition rates are slower than surfaces [187], subsoil C stocks are nonetheless more dynamic than previously proposed [229]. Therefore, both layers will affect global C flux, and should be incorporated as distinct units in models. Indeed, development of such a multi-layer model led researchers to conclude that treating the soil profile as a homogeneous unit actually vastly overestimates rates of C flux as a function of soil warming [187].

Developing more accurate multi-layer models requires that all pertinent differences between soil layers be taken into account. For instance, it needs to be better understood how C dynamics differ by depth [7], particularly as climate change is anticipated to exacerbate the frequency and severity of soil disturbances. Other parameters that should be incorporated into models include how SOM diffusion rates and microbe-substrate interaction frequencies decrease over depth [7,157], the nature of C downward translocation [187], and how microbial networks and synergistic interactions mediate SOM breakdown across soil depths [154]. In addition, information is needed about how the C is subsequently stabilized or decomposed in deeper soils [180]—especially as many of these parameters are strong determinants on C dynamics in subsoil but are less important in surface soils. Another problem with current models is the overestimation of the importance of chemical recalcitrance and underestimation of other SOC stabilization mechanisms [195,230], which is partly due to the lack of empirical data on soil aggregation [231]. Updating soil C modeling approaches by taking into account how SOC is processed by microorganisms in subsoils is thus a necessary step for making accurate predictions of C fluxes.

## Figures and Tables

**Figure 1 microorganisms-10-00540-f001:**
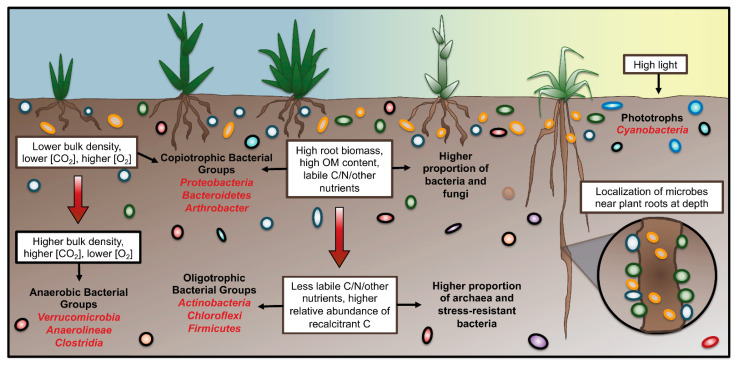
The effects of soil depth on soil properties and microbial groups. With depth, there is greater compaction and higher bulk density that decrease the oxygenation of subsoils, as well as prevent carbon dioxide from exchanging with the atmosphere. Together, these lead to the proliferation of anaerobic microbial groups in subsoils (**left**). At surface levels, there are higher contents of nutrients and labile C from sources such as decomposing plant OM and root exudates. These favor high microbial biomass and diversity, especially for copiotrophic bacterial groups, as well as overall levels of fungi and bacteria (**top middle**). However, in subsoils, the contents of labile C and nutrients go down due to fewer inputs from plants and other sources, leading to more nutrient-poor and extreme conditions which favor archaea and stress-resistant bacteria. Furthermore, much of the available C is in recalcitrant forms, which favors oligotrophic bacteria (**bottom middle**). The few fresh inputs of labile C tend to be around plant roots or soil cracks, so bacteria tend to be localized to these areas (**bottom right**). In addition, high light availability at the soil surface is conducive to phototrophic groups (**top right**).

## Data Availability

Not applicable.

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
