# Peer review of "Trends in Microbial Community Composition and Function by Soil Depth"

_microorganisms, 2022, doi:10.3390/microorganisms10030540_

Round 1

Reviewer 1 Report

Dear editor, dear authors!
Thank you very much for the opportunity to read your manuscript. In this case, for me, this is that rare moment when I have nothing special to add to the work done by the authors. The review looks very detailed and touches on the main aspects of the microbial role in soil formation. In general, I believe that this manuscript can be accepted for publication in its current form. As a suggestion, I could invite the authors to describe the role of microbial exopolysaccharides in the process of mineral weathering and the formation of soil aggregates.

Author Response

We thank the reviewer for these kind comments! Furthermore, per their suggestion we have incorporated a brief section of microbial EPS, how they change with soil depth, and their relationship to mineral weathering and aggregate formation (lines 113-130 in the revised manuscript).

Reviewer 2 Report

The article is extremely well documented and of great interest. Review articles are in general very useful but also hard to realize. The authors were able to synthesize very well the vast material extracted from the specialty literature. Indeed, the subject of soil and its resident microbiome that change with depth is not well studied. In this way, a synthesis of the investigations focused on this topic is more than welcomed. 

My main observation is that the article does not respect the common structure of an article (even though it is a review type article). For example, the Materials and Methods as well as the Discussion chapters are missing. 

Please mention how you have found the studied articles in the Materials and Methods section: where have you searched for them, on Web Science, Science Direct, SpringerLink Google Scholar? What keywords have you used? Also, what criteria did you use for searching articles?

I assume that the Results chapter is at page 2, above the first point, Soil Biogeochemical Properties and Depth. Please complete it. You can write Results and Discussion if the magazine’s format allows it. If not, the discussions mut be separated in a distinct chapter.

How did you choose the sub-chapters and is there a connection between them? Maybe a graph would be interesting. 

Personally, I think that the last point (Incorporating Subsoils into Climate Models) has no place here as the current article should be only about microbial community compositions and function by soil depth. The climate model issue is a different problem and should be approached in a different article. Actually, this sub-chapter is very short, another reason for eliminating it. 

Author Response

We thank this reviewer for their feedback! They were concerned with the methodology behind selecting papers to include in this review; however, we believe that that approach is principally suited for a systematic review, rather than a literature review. We have formatted this review similar to others published previously in Microorganisms (for one example, see "Gut microbiota, macrophages and diet: an intriguing new triangle in intestinal fibrosis", by Amamou et al., published 2/22/2022 in Microorganisms).

The reviewer's other principal comment was concerning the final section on climate modeling and its lack of direct relevance to the remainder of the review. With that in mind, we have eliminated this topic as its own section; instead we have incorporated the gist of this section into the 'conclusions and future directions' as a way to contextualize the importance of understanding the effects of soil depth on the microbiome.